# Learning Latent Semantic Representation from Pre-defined Generative Model

## Abstract

Learning representations of data is an important issue in machine learning. Though GAN has led to significant improvements in the data representations, it still has several problems such as unstable training, hidden manifold of data, and huge computational overhead. GAN tends to produce the data simply without any information about the manifold of the data, which hinders from controlling desired features to generate. Moreover, most of GAN's have a large size of manifold, resulting in poor scalability. In this paper, we propose a novel GAN to control the latent semantic representation, called LSC-GAN, which allows us to produce desired data to generate and learns a representation of the data efficiently. Unlike the conventional GAN models with hidden distribution of latent space, we define the distributions explicitly in advance that are trained to generate the data based on the corresponding features by inputting the latent variables that follow the distribution. As the larger scale of latent space caused by deploying various distributions in one latent space makes training unstable while maintaining the dimension of latent space, we need to separate the process of defining the distributions explicitly and operation of generation. We prove that a VAE is proper for the former and modify a loss function of VAE to map the data into the pre-defined latent space so as to locate the reconstructed data as close to the input data according to its characteristics. Moreover, we add the KL divergence to the loss function of LSC-GAN to include this process. The decoder of VAE, which generates the data with the corresponding features from the pre-defined latent space, is used as the generator of the LSC-GAN. Several experiments on the CelebA dataset are conducted to verify the usefulness of the proposed method to generate desired data stably and efficiently, achieving a high compression ratio that can hold about 24 pixels of information in each dimension of latent space. Besides, our model learns the reverse of features such as not laughing (rather frowning) only with data of ordinary and smiling facial expression.

## 1 Introduction

Developing generative model is a crucial issue in artificial intelligence. Creativity was a human proprietary, but many recent studies have attempted to make machines to mimic it. There has been an extensive research on generating data and one of them, generative adversarial network (GAN), has led to significant achievements, which might be helpful to deep learning model because, in general, lots of data result in good performance (LeCun et al., 2015). Many approaches to creating data as better quality as possible have been studied: for example, variational auto-encoder (VAE) (Kingma & Welling, 2013) and GAN (Goodfellow et al., 2014). The former constructs an explicit density, resulting in an explicit likelihood which can be maximized, and the latter constructs an implicit density (Goodfellow, 2016). Both can generate data from manifold which is hidden to us so that we cannot control the kind of data that we generate.

Because it is costly to structure data manually, we need not only data generation but also automatically structuring data. Generative models produce only data from latent variable without any other information so that we cannot control what we want to generate. To cope with this problem, the previous research generated data first and found distributions of features on latent space by investigating the model with data, since the manifold of data is hidden in generative models. This latent space is deceptive for finding an area which represents a specific feature of our interest; it would

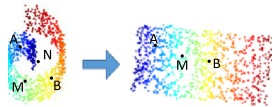

Figure 1: Examples of the manifold. Left: a complex manifold which can be seen in general models, Right: a relatively simple manifold in the proposed model. The midpoint M of A and B can be easily calculated in the right manifold, but not in the left one. The midpoint of A and B is computed as N in the left manifold, which is incorrect.

take a long time even if we can find that area. Besides, in the most of research, generative models had a large latent space, resulting in a low compression rate which leads to poor scalability. To work out these problems, we propose a model which can generate the data whose type is what we want and learn a representation of data with a higher compression rate, as well. Our model is based on VAE and GAN. We pre-define distributions corresponding to each feature and modify the loss function of VAE so as to generate the data from the latent variable which follows the specific distribution according to its features. However, this method makes the latent space to become a more complex multimodal distribution which contains many distributions, resulting in an instability in training the LSC-GAN. We prove that this problem can be solved and even made more efficiently by using an auto-encoder model with the theorem in Section 3. Although the proposed model compresses the data into small manifold, it is well-defined with Euclidean distance as shown in Fig. 1, which compares the manifolds in general models and in our model. The distance can be calculated with Euclidean distance in adjacent points but not in far points at the left manifold in Fig. 1. However, in the right manifold, we can calculate the distance between points regardless of the distance of them, where we can recognize the manifold more easily as shown in the left side. Thanks to a relatively simple manifold, it can produce neutral features regardless of their location in latent space, so that all features can be said as independent to each other. Our main contribution is summarized as follows.

- We propose a method to improve the stability of a LSC-GAN with LSC-VAE by performing the weight initialization, and prove it theoretically.

- We achieve conditional generation without additional parameters by controlling the latent space itself, rather than adding additional inputs like the existing model for condition generation.

- We propose a novel model that automatically learns the ability to process data continuously through latent space control.

- Finally, we achieve an efficient compression rate with LSC-GAN based on weight initialization of LSC-VAE.

The rest of the paper is organized as follows. Section 2 reviews the related works and the proposed LSC-GAN model is illustrated in Section 3. In Section 4, we evaluate the performance of the proposed method with some generated data. The conclusion and discussion are presented in Section 5.

## 2 RELATED WORKS

Many research works have been conducted to generate data such as text, grammar, and images (Yang et al., 2017; Kusner et al., 2017; Denton et al., 2015). We divide the approaches for data generation into three categories: only generation, conditioned generation, and transforming data to have different features.

Several researchers proposed generative models of VAE and GAN (Kingma & Welling, 2013; Goodfellow et al., 2014). These are basis of the generative models. Both use maximum likelihood approach, but they have different policies to construct density: explicitly and implicitly. There are lots of variations of these models. Radford et al. (2015) constructed deep convolutional GAN (DCGAN) with convolutional neural networks (CNN) for improving the performance with the fact that CNN had been huge adoption in computer vision applications. Zhao et al. (2016) introduced energy-based GAN (EBGAN) using autoencoder in discriminator. Kim et al. (2017a; 2018a;b) proposed transferred encoder-decoder GAN (TED-GAN) for stabilizing process of training GAN and used it to classify

the data. These studies focused on high productivity in generation so that they could not control the type of generated data.

Recently, some researchers began to set conditions on the data they generate. Sohn et al. (2015) and Walker et al. (2016) inputted data and conditions together into VAE and generated data whose type is what they want, called conditional VAE (CVAE). van den Oord et al. (2017) set discrete embedding space for generating a specific data using vector quantized variational auto-encoder (VQ-VAE), but because of discrete space, they could not control latent space continuously. Larsen et al. (2015) used both VAE and GAN in one generative model. As they just mixed two models and did not analyzed a latent space, so that the manifold of data was hidden to us. To generate image with a specific feature, they extracted a visual attribute vector which is a mean of vector in latent space. Mirza & Osindero (2014) inputted not only data but also conditions into GAN to create data that we want, called conditional GAN (CGAN). Chen et al. (2016) used mutual information for inducing latent codes (InfoGAN) and Nguyen et al. (2017) added a condition network that tells the generator what to generate (PPGN) . These two models needed an additional input to generate the type of data we want. These studies make us to generate data with condition, but we still do not know about latent space and it is hard to find the location of a specific feature in the latent space. Therefore, we propose a model that learns to generate concrete features that we want from the latent space determined when LSC-VAE is trained.

Some studies attempted to transfer the given data to others which have different features or even in different domain. Tran et al. (2017) proposed disentangled representation learning GAN (DRGAN) for pose-invariant face recognition. Reed et al. (2016a;b) tried matching latent space of text and images and finally they translated text to image. Zhang et al. (2017) also translated text to image and generated photo-realistic images conditioned on text by stacking models (StackGAN). Zhu et al. (2017) and Kim et al. (2017b) discovered cross-domain relations with CycleGAN and DiscoGAN. They can translate art style, face features, and bags to shoes. While other models could only do one conversion task, Choi et al. (2017) proposed StarGAN that could do multiple translation tasks with one model. These studies have been conducted to transform the data into those in other domains. However, they could not generate new data without input data. In addition, the size of latent space of most of them was too large. We aim to generate conditioned data even with a small size of latent space.

## 3 THE PROPOSED METHOD

In this section, we present a method to generate the data with the corresponding characteristics by inputting the latent variable which follows the specific distribution in latent space. As the instability caused by the larger scale of latent space in this process, we use the modified VAE, called LSC-VAE[1]. As shown in Fig. 2(a), we train the LSC-VAE with $L_{prior}$ for the data to be projected by the encoder into the desired position in the latent space according to the characteristics of the data. The trained decoder of the LSC-VAE is used as a generator of LSC-GAN so that the LSC-GAN generates the data with the corresponding features by using latent variables sampled from a specific distribution.

The proposed model is divided into two phases: initializing latent space (Fig. 2(a)) and generating data (Fig. 2(b)). In the first phase, latent semantic controlling VAE (LSC-VAE) is trained to project data into a specific location of latent space according to its features, and it learns to reconstruct data which is compressed. The decoder of LSC-VAE is used in the generator ($G$) of LSC-GAN in the second phase. $G$ and discriminator ($D$) are trained simultaneously so that $G$ can produce data similar to real data as much as possible and that $D$ can distinguish the real from the fake. The architecture of the generation process is shown in Fig. 2(b).

### 3.1 INITIALIZING THE LATENT SPACE WITH LSC-VAE

Auto-encoder has been traditionally used to represent manifold without supervision. In particular, VAE, one type of auto-encoders, is one of the most popular approaches to unsupervised learning of complicated distributions. Since any supervision is not in training process, the manifold constructed is hidden to us. As we mentioned in Section 1, this is usually too complex to generate the conditioned

---

[1]Intuitively, as opposed to GAN, VAE constructs the distribution of data explicitly, resulting in efficient training of LSC-GAN. We deal with this theoretically in Section 3.2.

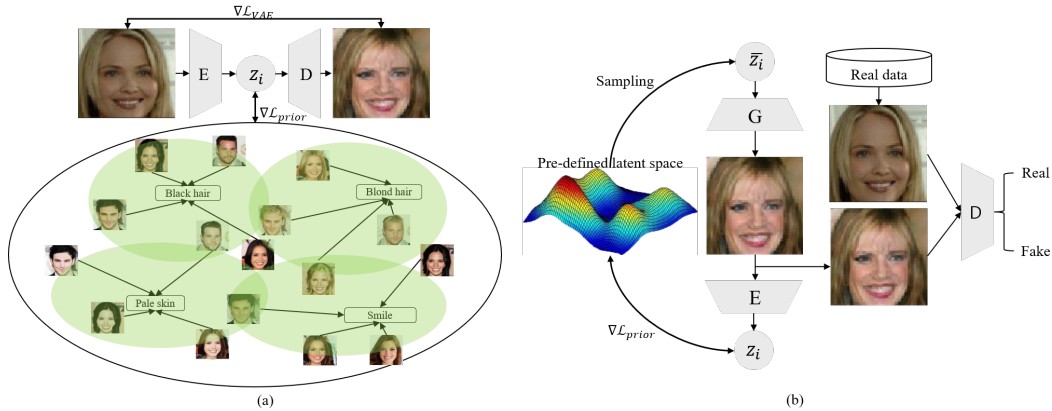

Figure 2: (a) The process of pre-defining a latent space. The LSC-VAE is trained to project the data into the appropriate position on latent space. (b) Generating process of the proposed method. The latent space is pre-defined in the process of (a).

data. Therefore, we allow LSC-VAE to learn a representation of data with supervision. It compresses data into a particular place on latent space according to its features. The proposed model consists of two modules that encode a data $x_i$ to a latent representation $z_i$ and decode it back to the data space, respectively.

$$z_i \sim Enc(x_i) = Q(z_i \mid x_i) \tag{1}$$
$$\tilde{x}_i \sim Dec(z_i) = P(x_i \mid z_i) \tag{2}$$

Index $i$ means a feature which is included in data $x$ and latent space $z$. The encoder is regularized by imposing a prior over the latent distribution $P(z)$. In general, $z \sim N(0, I)$ is chosen, but we choose $z_i \sim \mathcal{N}(\mu_i, I)$ for controlling latent space. In addition, if we want to produce data which has multiple features $i, j$, we generate data from $z_{ij} \sim \mathcal{N}(\mu_i + \mu_j, I)^2$. The loss function of LSC-VAE is as follows.

$$\mathcal{L}_{LSC-VAE} = -\mathbb{E}_{z_i \sim Q(z_i|x_i)}[logP(x_i \mid z_i)] + \mathcal{D}_{KL}[Q(z_i \mid x_i) \parallel P(z_i)]$$
$$= \mathcal{L}_{VAE} + \mathcal{L}_{prior} \tag{3}$$

where $D_{KL}$ is the Kullback-Leibler divergence. The first term of equation 3 is related to reconstruction error and the second term is related to appropriate projection of data to the latent space. For example, when LSC-VAE projects the data with $i-$ and $j-$features into the latent space, it is trained to map the data into the pre-defined latent space ($\mathcal{N}(\mu_i + \mu_j, I)$) with $L_{prior}$ in equation 3 so as to locate the reconstructed data as similar to the input data according to its characteristics using $L_{VAE}$. Therefore, LSC-VAE can be used in initializing GAN and it is demonstrated that LSC-VAE is valid and efficient for LSC-GAN in the next section.

## 3.2 GENERATING DATA WITH LSC-GAN

GAN has led to significant improvements in data generation (Goodfellow et al., 2014). The basic training process of GAN is to adversely interact and simultaneously train $G$ and $D$. However, because the original GAN has a critical problem, unstable process of training (Radford et al., 2015), the least squares GAN (LS-GAN) is proposed to reduce the gap between the distributions of real data and fake data by Mao et al. (2017). Equation 4 shows the objective function of the LS-GAN. $p_{data}$ is the probability distribution of the real data. $G(z)$ is generated data from a probability distribution $p_z$, and it is distinguished from the real by $D$.

---

[2]More precisely, $\mathcal{N}(\mu_i + \mu_j, 2I)$ is correct but calculated as $\mathcal{N}(\mu_i + \mu_j, I)$ for convenience in computation and scalability.

$$\min_D \mathbb{E}_{x \sim p_{data}(x)}[(D(x) - 1)^2] + \mathbb{E}_{z \sim p_z(z)}[(D(G(z)))^2]$$
$$\min_G \mathbb{E}_{z \sim p_z(z)}[D(G(z)) - 1)^2] \tag{4}$$

The main differences of the proposed model with VAE-GAN and LS-GAN is that LSC-GAN is based on LSC-VAE for initializing a latent space to control it. To produce the type of data we want, we just input latent variable $z_i \sim \mathcal{N}(\mu_i, I)$ to $G$, if the data has $i-$feature. Besides, we add the encoder of LSC-VAE into LSC-GAN to make sure that the generated data actually have the desired features. The encoder projects back to latent space so as to be trained to minimize the difference between latent space where data is generated and the space where the compressed data is projected. Equation 5 is about loss of $D$ and loss of encoder and $G$.

$$\min_D \mathbb{E}_{x_i \sim p_{data}(x_i)}[(D(x_i) - 1)^2] + \mathbb{E}_{z_i \sim p_z(z_i)}[(D(G(z_i)))^2]$$
$$\min_{Q,G} \mathbb{E}_{z_i \sim p_z(z_i)}[D(G(z_i)) - 1)^2] + \mathcal{D}_{KL}[Q(z_i \mid G(z_i)) \| \mathcal{N}(\mu_i, I)] \tag{5}$$

### 3.2.1 PRE-TRAINED GENERATOR

Since the original GAN has disadvantage that the generated data are insensible because of the unstable learning process of the $G$, we pre-train $G$ with decoder of LSC-VAE. The goal of the learning process of generating data of $G$ is the same as equation 6 from equation 5, and it is equivalent to that of equation 7. However, it is not efficient to pre-train the $G$, because it depends on the parameters of the $D$. Therefore, we change this equation to equation 8 again, and it is represented only by the parameters of $G$. In this paper, to train the $G$ with equation 8, we use the decoder of LSC-VAE, which is trained by using $Dec(Enc(x)) \approx x$. The result of LSC-VAE is that $\mid p_{data}^{LSC-VAE} - p_G^{LSC-VAE}) \mid \leq \mid p_{data} - p_G \mid$ so that it can reach a goal of GAN ($p_{data} \approx p_G$) stably, which is proved by Theorem 1 and 2.

$$\min_G (1 - D(G(z_i)))^2 \tag{6}$$
$$\Leftrightarrow D(G(z_i)) \approx 1 \tag{7}$$
$$\Leftrightarrow G(z_i) \approx x \in \mathcal{X}_i, \tag{8}$$

where $\mathcal{X}_i$ is real dataset with $i-$feature.

### 3.2.2 VALIDITY AND EFFICIENCY OF LSC-VAE

From the game theory point of view, the GAN converges to the optimal point when $G$ and $D$ reach the Nash equilibrium. In this section, let $p_G$ be the probability distribution of data created from $G$. We show that if $G(z) \approx x$, i.e., $p_{data} \approx p_G$, the GAN reaches the Nash equilibrium. We define $J(D, G) = \int_x p_{data}(x)D(x)dx + \int_z p_z(z)(1 - D(G(z)))dz$ and $K(D, G) = \int_x p_{data}(x)(1 - D(x))dx + \int_z D(G(z))dz^3$. We train $G$ and $D$ to minimize $J(D, G)$ and $K(D, G)$ for each. Then, we can define the Nash equilibrium of the LSC-GAN as a state that satisfies equation equation 9 and equation equation 10. Fully trained $G$ and $D$ are denoted by $G^*$ and $D^*$, respectively.

$$J(D^*, G^*) \leq J(D^*, G) \forall G \tag{9}$$
$$K(D^*, G^*) \leq K(D, G^*) \forall D \tag{10}$$

**Theorem 1.** If $p_{data} \approx p_G$ almost everywhere, then the Nash equilibrium of the LSC-GAN is reached.

Before proving this theorem, we need to prove the following two lemmas.

**Lemma 1.** $J(D^*, G)$ reaches a minimum when $p_{data}(x) \leq p_G(x)$ for almost every $x$.

---

[3]Since $(1_{p_{data}(x)>p_G(x)})^2 = 1_{p_{data}(x)>p_G(x)}$ and $(1 - 1_{p_{data}(x)>p_G(x)})^2 = (1 - 1_{p_{data}(x)>p_G(x)})$, we eliminate the square.

**Lemma 2.** $K(D, G^*)$ reaches a minimum when $p_{data}(x) \geq p_{G^*}(x)$ for almost every $x$.

The proof of Lemma 1 and 2 were discussed by Kim et al. (Kim et al., 2018b). We assume that $p_{data} \approx p_G$. From Lemma 1 and Lemma 2, if $p_{data} \approx p_G$, then $J(D, G)$ and $K(D, G)$ both reach minima. Therefore, the proposed GAN reaches the Nash equilibrium and converges to optimal points. By theorem 1, GAN converges when $p_d \approx p_g$, and it is done to some extent by the modified VAE, i.e. $| p_d^{LSC-VAE} - p_g^{LSC-VAE} | \leq | p_d - p_g |$ since the one goal of modified VAE is $P(x \mid Q(z \mid x)) \approx x$. Therefore, the proposed method is useful to initialize the weight of the generative model. However, it shows only validity of using VAE when learning GAN. We prove that it is also efficient by proving theorem 2. Assume that a model f is well-trained if and only if $\nabla L_f \approx 0$, where $L_f$ is the loss function of $f$.

**Theorem 2.** Let $en_k, de_k$ be $k$ epoch-trained encoder and decoder whose goal is $de_k(en_k(x)) \approx x \in \mathcal{X}$. $D$ and $de$'s are linear functions.[4] Let $\mathcal{L}_G^k = \mathbb{E}_{z \sim Q(z|x)}[(D(de_k(z)) - 1)^2]$, then $\nabla \mathcal{L}_f \to 0$ as $k \to \infty$.

*Proof.* Notice that the derivative is unique, and a derivative of linear function is itself. Since $en$ and $de$ are trained with $L_{VAE}$ and $L_{prior}$, the following statement is satisfied.

$$\forall \epsilon \geq 0, \exists N \in \mathbb{N} s.t. \forall k \geq N, \| de_k(z) - x \| < \epsilon \tag{11}$$

By differentiating the formula,

$$\nabla \mathcal{L}_G^k \approx 0 \text{ as } k \to \infty \tag{12}$$
$$\Leftrightarrow [p(z)(D(de_k(z)) - 1)\mathbb{D}D(de_k(z))](z) \to 0 \text{ as } k \to \infty$$

where $z \sim Q(z \mid x)$. Since the derivative of linear function is itself, it derives to

$$\Leftrightarrow p(z)(D \cdot de_k(z) - 1)D \cdot de_k(z) \to 0 \text{ as } k \tag{13}$$

With the fact that $D(x) = 1, \forall x \in \mathcal{X}$ and equation 11, it finally derives to

$$p(z)(D(x) - 1)D(x) \to p(z) \cdot 0 \cdot 1 \text{ as } k \to \infty \tag{14}$$

By theorem 1 and 2, the proposed learning process is valid and efficient.

## 4 EXPERIMENTS

### 4.1 DATASET AND EXPERIMENTAL SETTING

To verify the performance of the proposed model, we use the celebA dataset (Liu et al., 2015). It is a large-scale face attributes dataset. We crop the initial 178×218 size to 138×138, and resize them as 64×64. We use 162,769 images in celebA and 14 attributes: black hair, blond hair, gray hair, male, female, smile, mouth slightly open, young, narrow eyes, bags under eyes, mustache, eyeglasses, pale skin, and chubby. We assign 20 dimensions to each feature and set mean of the $i^{th}$-20 dimensions as 1. For example, if an image has i-feature, the elements of $i * 20^{th}$ to $(i + 1) * 20^{th}$ of the image's latent variable are 1 in average and 0 in the remainder and we denote that latent variable as $n_i$.

### 4.2 GENERATED IMAGES

As shown in Fig. 3, we generate images from a specific latent space by using LSC-GAN. The images in the first column are generated to have 'female' and 'blond hair' features. We confirm that the

---

[4]In fact, it is enough that $D$ does not satisfy $\mathbb{D}D = D^a$, where $a \leq -1$.

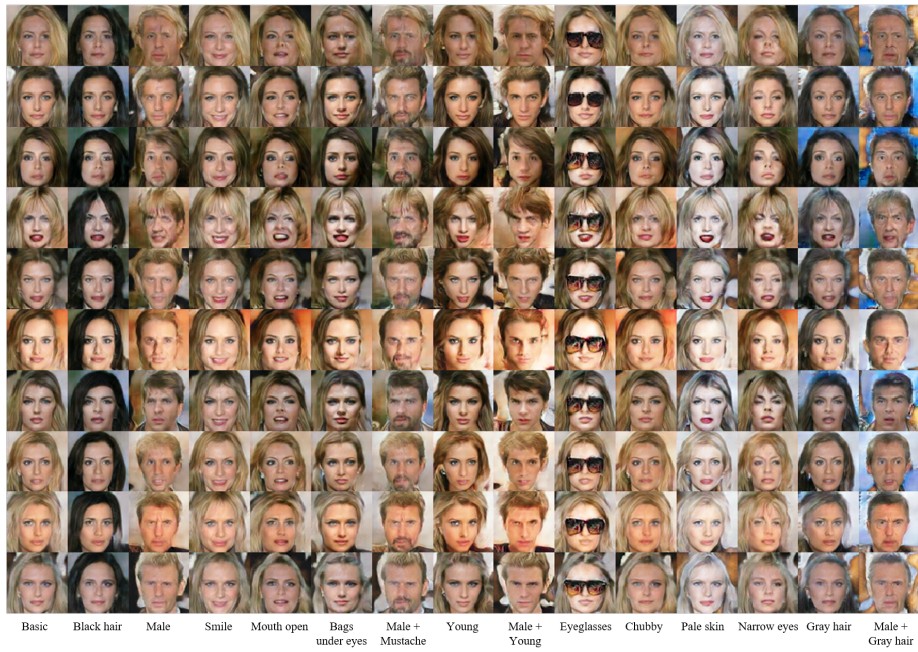

Figure 3: The generated images. The images are shown in each column according to the features below the columns.

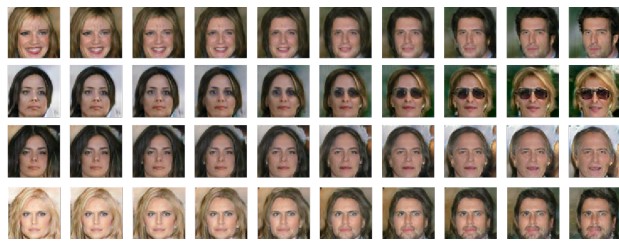

Figure 4: Interpolation between a series between images in leftmost and rightmost columns.

condition works well. The images in the remaining columns are transformed using equation 15 for the features listed below. For example, if we generate an image $x_i$ which has $i-$feature from the latent variable $z_i$, we add $n_j$ to add $j-$feature into the image.

$$x_{ij} = G(z_i + n_j) \tag{15}$$

where $x_{ij}$ is an image which has $i-$ and $j-$features, and $z_i$ is the latent variable for $i-$feature. To show that the proposed model does not simply memorize data but understand features of data and generate them, we generate images from a series between two random images as in DCGAN. As shown in Fig. 4, the change between images is natural so that we can say that the latent space of LSC-GAN is a manifold. Besides, the images in the middle column have both features of images in leftmost and rightmost, resulting in more simple manifold as shown in Fig. 1.

Unlike other GAN models, the LSC-GAN fully understands features of data so as to generate data including inverse-feature. We only train the model about the presence of the 'pale skin' and 'smile' features, but the model also learned about the reverse of 'pale skin' and 'smile' automatically as shown in the fourth and the ninth column of Fig. 5. Besides, if we assign a value of 2 rather than 1 to the average of latent variable which is related to 'mustache', we can see that more mustaches are created in the last column in Fig. 5. Therefore, our model can automatically infer and generate the

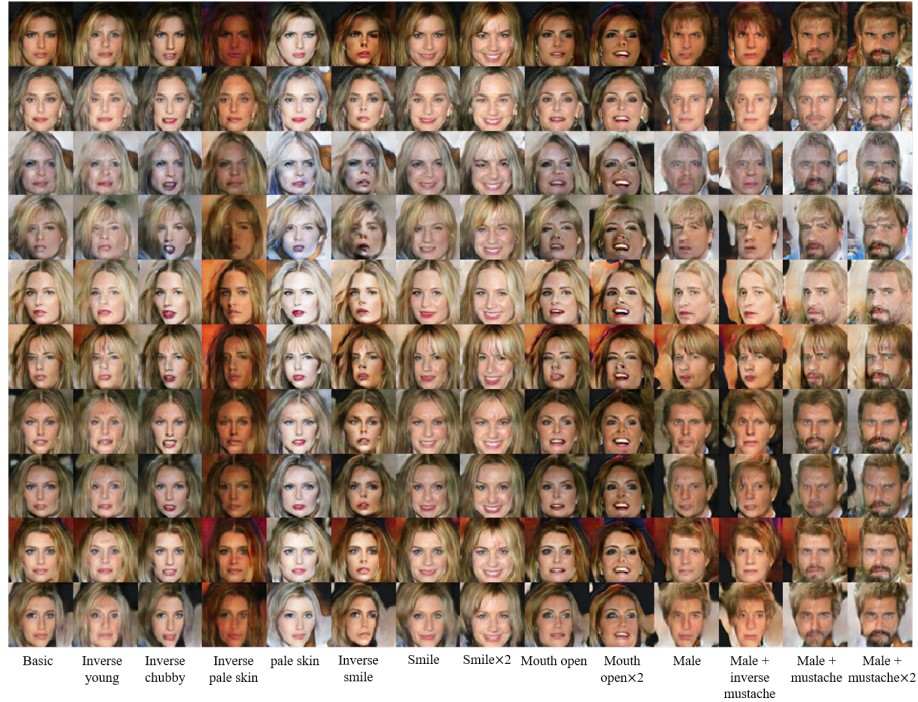

| Basic | Inverse young | Inverse chubby | Inverse pale skin | pale skin | Inverse smile | Smile | Smile×2 | Mouth open | Mouth open×2 | Male | Male + inverse mustache | Male + mustache | Male + mustache×2 |

Figure 5: The result of generating inverse-features. Our proposed model automatically learns inverse features such as dark skin (inverse of 'pale skin') and frown (inverse of 'smile').

data with inverse-feature that do not exist in the dataset. This shows that the proposed model has the ability to deduce a negative feature by itself although only positive features are used in training

To verify the proposed model, we conduct subjective test about the quality of the generated data. We generate data by using DCGAN, EBGAN, and the proposed GAN. We randomly choose 25 generated data for each model. We perform the subjective test on 30 subjects and ask them to evaluate the quality of the generated data in 5 ways: very low, low, medium, high, and very high. We collect the results of 750 questionnaires, which are the evaluated result of 25 generated images by 30 subjects, and summarize them in Table 1. We score 1,2,3,4, and 5 points for each evaluation result which is shown in the last column in Table 1.

Table 1: The results of subjective test about the quality of the generated data by DCGAN, EBGAN, and the proposed model.

| Model | DCGAN | EBGAN | Ours |
|---|---|---|---|
| Very low (1) | 38.8% | 18.2% | 3.6% |
| Low (2) | 38.5% | 37.6% | 20.8% |
| Medium (3) | 15.3% | 28.9% | 36.7% |
| High (4) | 4.3% | 11.6% | 24.8% |
| Very high (5) | 3.1% | 3.6% | 14.1% |
| Score | 1.943 | 2.447 | **3.251** |

### 4.3 COMPRESSION RATE

Our model not only generates images according to input conditions, but also compress efficiently. We calculate the compression rate with rate= $size_{inputdata}/size_{bottleneck}/\#classes$. As shown in Table 2, our proposed model has the best compression rate compared to others. This proves experimentally that LSC-VAE, theoretically proven with theorems 1 and 2, has been helpful in initializing the weights of the LSC-GAN, and it can achieve good performance even in small latent spaces.

Table 2: The compression rate of models with 14 classes.

| Model | U-NET | VQ-VAE | DiscoGAN | CycleGAN | StarGAN | LSC-GAN |
|---|---|---|---|---|---|---|
| Compression rate | 0.504/14 | 48.006/14 | 40.922/14 | 0.378/14 | 0.188 | **24.546** |

## 5 CONCLUSION

In this paper, we address some of significant issues in generative models: unstable training, hidden manifold of data, and extensive hardware resource. To generate a data whose type is what we want, we propose a novel model LSC-GAN which can control a latent space to generate the data that we want. To deal with a larger scale of latent space cause by deploying various distributions in one latent space, we use the LSC-VAE and theoretically prove that it is a proper method. Also, we confirm that the proposed model can generate data which we want by controlling the latent space. Unlike the existing generative model, the proposed model deals with features continuously, not discretely and compresses the data efficiently.

Based on the present findings, we hope to extend LSC-GAN to more various datasets such as ImageNet or voice dataset. In future work, we plan to conduct more experiments with various parameters to confirm the stability of model. We will also experiment by reducing the dimension of the latent space to verify that the proposed model is efficient. Besides, since the encoder can project the data to the latent space according to the features inherent in data, it could be used as a classifier.

ACKNOWLEDGMENTS

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
