# OpenReview forum: "Learning Latent Semantic Representation from Pre-defined Generative Model"
_ICLR.cc/2019/Conference_

### Official Review · AnonReviewer1 · 2018-11-02
**Really confusing**

**Rating:** 4
**Confidence:** 3

**Review:**

This paper proposes to learn a manifold of deep generative models using a pre-trained VAE. To generate samples with desired features, this paper proposes to learn an embedding of each feature in the hidden space using VAE. Then the learned hidden space is used to train a GAN.

However, the method in this paper and main contributions are not clearly represented. I can hardly understand the motivation of this paper. In the introduction part, this paper mentions “large scale of latent space” lots of times, but does not make it clear that why a large latent space hinders the deep generative models. In Fig.1, it demonstrates that for some manifold, L2 distance cannot be applied directly. However, for most DGMs, the hidden space is defined in Euclid Space, and L2 distance is a valid distance for them.

In Sec. 3, the method is not presented clearly and the notation is confusing. In Sec. 3.2.1, Eqn (8) is not an objective function and it is confusing how to optimize the generator using it. In Sec. 3.2.2, the notation is really confusing and I can hardly understand the proof the Theorem 2.

The experimental results are not solid where no well-known metrics, such as Inception Score, FID, are used to evaluate the generated samples. For compression rate, the size of bottleneck has not been mentioned above, and the experimental setting of each baseline is not ignored which makes the experimental results incomparable.

Overall, this paper is not a qualified paper for ICLR.

---

### Official Review · AnonReviewer3 · 2018-11-02
**Poorly written draft with minimal technical novelty**

**Rating:** 3
**Confidence:** 5

**Review:**

The paper proposes a generative model that combines VAE and GAN. The main idea of the paper is to replace the standard normal distribution used in VAE with a normal distribution centered at a feature representation of the input image. In other words, the prior distribution is data adaptive. The paper compares the proposed generative model to DCGAN and EBGAN for image generation quality using the CelebA dataset and reports better human preference score.

Overall, the paper is poorly written with incorrect technical descriptions and vague expositions.  The two baselines (DCGAN and EBGAN) are also quite out-dated. Beating these two baselines are insignificant, particularly there are GAN methods that  can generate high quality images without an encoder. For example, the Progressive GAN by Terro et. al. (ICLR 2018), SNGAN by Miyato et. al.(ICLR 2018), and GAN with zero-center gradient penalty by Mescheder et. al. (ICML 2018). The paper also fails to give a literature overview of effort in combining VAE and GAN. For example, Zhiting et. al. ICLR 2018 and Liu et. al. NIPS 2017.

Technical errors

- In the related works section, the paper states that VAEs and GANs are both based on maximum likelihood. This statement is incorrect as GANs are based on distribution matching.

Vague exposition

- In Section 2, the paper states that "Larsen et al. (2015) used both VAE and GAN in one generative model. As they just mixed two models and did not analyzed a latent space, so that the manifold of data was hidden to us." Isn't the data manifold in this case a multivariate Gaussian distribution. The paper fails to explain what it means by the sentence.

- In Section 3.1, the paper states that "Since any supervision is not in training process, the manifold constructed is hidden to us." Again, the reviewer fails to understand what the paper means.

- In Section 3.2.1, the paper states that "it is not efficient to pre-train the G , because it depends on the parameters of the D." This sentence is confusing. Isn't pretraining just meaning using a pretrained decoder weight to initialize G?

---

### Official Review · AnonReviewer2 · 2018-11-03
**I think it's an interesting approach to an interesting problem.  I am not familiar with other SOTA results, but visually the results are not that compelling. The explanation of the method should be clarified.**

**Rating:** 5
**Confidence:** 2

**Review:**

* Pros
- addresses an interesting problem
- gives a nice approach to the problem
- attempts to give some theoretical justification for the approach

* Cons
- I generally understand the approach, but details were not clear to me (specifics given below)
- Sections 3.2.1 and 3.2.2 (the theoretical section), I found particuarly hard to follow.
- The visual results are not particularly compelling, tbhough I suppose the panel liked them better than the competitor methods (Table 1).  For example "inverse pale skin" and "pale skin" in figure 5 do not convince me that the model understands skin. The skin and background seem to be changing colors together.  Might be worth including examples of the competitor approaches to show that they are even worse.

* Comments and Questions
- Throughout, you seem to assume binary-valued features, without ever explicitly stating this.  Would be helpful to state explicitly.
- Would be useful to specify the codomain of the discriminator D(x) -- from the objective function, seems to be a value in (0,1).
- In Section 3.2, you say: "Besides, we add the encoder for LSC-VAE into LSC-GAN to make sure that the generated data actually have the desired features.  The encoder projects back to latent space so as to be trained to minimize the difference between latent space where data is generated and the space where the compressed data is projected."  This seems like a fundamental change to training, much more than just initializing a GAN with a VAE-GAN. I think this should be elaborated on.  For example, what happens if you don't include the term with the encoder in (5).  Moreover, what goes wrong if you just try to train everything jointly using (5) without the VAE-GAN initialization step?
- I have a very difficult time understanding 3.2.1, both the text and the equations.  e.g. what are the z_i's in (8)?  In 3.2.1, you say "we pre-train G with the decoder of LSC-VAE" -- this "pretraining" is what you refer to as initialization previously, I think?  Which seems also what section 3.1 is about?
- I think some clarification would be nice in Section 3.2.2.  The conclusion of the section is that the "proposed learning process is valid and efficient."       What do you mean by "valid" and "efficient"?  Perhaps you can explain that a bit more in words at the beginning of the section.  It's not entirely clear where your theory section connects to the objective function in (5). I don't really follow your argument for LSC_VAE being a good initializer in this section. In what sense is theorem 2 demonstrating "efficiency"?
- In Theorem 1, you write p_data \approx p_G.  I guess p_data is some unknown data generating distribution, rather than the empirical distribution of a training set?   I've also never seen \approx 0 used in formal mathematical statements and proofs especially when we're talking about \approx 0 at infinitely many points.  Can this be elaborated on?
- In the end of section 3 intro paragraph you say "The decoder of LSC-VAE is used in the generator (G) of LSC-GAN in the second phase." By "used" do you mean used as the initialization of the generator G, when we switch to the LSC_GAN training?  Seems like it, but could be made more clear.
- In Section 3, second paragraph, you say "In the first phase, LSC-VAE is trained to project data into a specific location of latent space according to its features".  It's not clear whether or not this "projection step" (which I guess is also called encoding step or the inference step depending on the context?) uses the explicit feature values in this step, or can only use the input (e.g. the image).  I really have the same question for the decoder/generator: does it explicitly use the feature values, or does it depend only on the latent variables?  My guess is that in both cases the feature values are not depended on directly, but I think this could be made more clear, one way or the other.
- how did results vary when you deviate from using 20 latent dimension per feature?
- You say "As shown in Fig. 4, the change between images is natural so that we can say that the latent space of LSC-GAN is a manifold." --- maybe a linear manifold?
- Footnote 2 on page 4: This is confusing.  You are [basically arbitrarily] defining the conditional distribution on the latent space for any feature setting.  How can any particular distribution be "correct".
- In the equations in (5), you're taking expectations over z_i, but don't you need to have an expectation over i (the feature assigments) as well?  Do you use the same feature distributions as you have in the training data?  Should be clarified.
- Also, in (5), the expectation over z_i applies to the first term, as well as the z_i in G(z_i) in the second KL divergence term, right?
- In VAE the encoder typically produces the parameters of the Q distribution on the latent space.  What distribution does Q have and how are you parameterizing it?  Indendent Gaussians on each coordinate, each with its own mean and standard deviation?  or what? If you are allowing the encoder to take the feature values as input (which I don't think you are, but am not entirely sure of), does the encoder have to learn the means for each feature setting, or are you explicitly building those feature-based offsets into the encoder?

---

### Meta-Review · Area_Chair1 · 2018-12-15
**Concerns about clarity and writing of the paper**

**Confidence:** 5
**Recommendation:** Reject

**Metareview:**

Reviewers have expressed concerns about clarity/writing of the paper and technical novelty, which the authors haven't responded to. The paper is not suitable for publication at ICLR.